# Diffusion Barrier Characteristics of WSiN Films

**Yung-I Chen [1,2]** , **Kuo-Hong Yeh [1]** , **Tzu-Yu Ou [3]** and **Li-Chun Chang [3,4,*]**

1    Department of Optoelectronics and Materials Technology, National Taiwan Ocean University,
     Keelung 202301, Taiwan; yichen@mail.ntou.edu.tw (Y.-I.C.); 11089008@email.ntou.edu.tw (K.-H.Y.)
2    Center of Excellence for Ocean Engineering, National Taiwan Ocean University, Keelung 202301, Taiwan
3    Department of Materials Engineering, Ming Chi University of Technology, New Taipei City 243303, Taiwan;
     ouziyu23@mail.mcut.edu.tw
4    Center for Plasma and Thin Film Technologies, Ming Chi University of Technology,
     New Taipei City 243303, Taiwan
*    Correspondence: lcchang@mail.mcut.edu.tw; Tel.: +886-2-2908-9899

**Abstract:** WSiN films were produced through hybrid pulse direct current/radio frequency magnetron co-sputtering and evaluated as diffusion barriers for Cu metallization. The Cu/WSiN/Si assemblies were annealed for 1 h in a vacuum at 500–900 °C. The structural stability and diffusion barrier performance of the WSiN films were explored through X-ray diffraction, Auger electron spectroscopy, and sheet resistance measurement. The results indicated that the Si content of WSiN films increased from 0 to 9 at.% as the power applied to the Si target was increased from 0 to 150 W. The as-deposited $W_{76}N_{24}$, $W_{68}Si_0N_{32}$, and $W_{63}Si_4N_{33}$ films formed a face-centered cubic $W_2N$ phase, whereas the as-deposited $W_{59}Si_9N_{32}$ film was near-amorphous. The lattice constants of crystalline WSiN films decreased after annealing. The sheet resistance of crystalline WSiN films exhibited a sharp increase as they were annealed at 800 °C, accompanied by the formation of a $Cu_3Si$ compound. The failure of the near-amorphous $W_{59}Si_9N_{32}$ barrier against Cu diffusion was observed when annealed at 900 °C.

**Keywords:** Cu metallization; $Cu_3Si$; diffusion barrier; WSiN

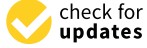

## 1. Introduction

Cu metallization has been applied as an interconnecting conductor in ultra-large-scale integrated circuits because of its high conductivity and excellent electromigration resistance [1–3]. The prohibition of Cu diffusion in Si and $SiO_2$ by introducing diffusion barriers has become the crucial subject for preventing the deterioration of device performance [4,5]. The formation of $Cu_3Si$ after annealing is an indicator used to evaluate the failure of the barrier, which is accompanied by an increase in the electrical resistance [6,7]. Distinct materials such as passive, stuffed, sacrificial, and amorphous barriers have been proposed as thin-film diffusion barriers [8]. Amorphous materials without grain boundaries exhibited excellent diffusion resistance [9,10] and were applied in the field of Cu metallization. Ta and W films play the role of diffusion barriers against Cu at 600 °C in $H_2$ for 1 h [11], which is attributed to their high melting temperatures and immiscibility with Cu [7,12]. Amorphous binary and ternary nitrides ($TaN_x$ [7,13], $WN_x$ [14,15], Ta–W–N [16], and W–Ti–N [17]) have been used to raise the barrier temperatures to 600–800 °C. High-entropy alloy nitride films, such as $(AlMoNbSiTaTiVZr)_{50}N_{50}$ [10], $(AlCrTaTiZr)N$ [18], and $(AlCrTaTiZrMo)N$ [19], with amorphous structures and large lattice distortions, have been applied as diffusion barriers [20]. The phase transformation of the aforementioned nitride films at elevated temperatures affected the performance of diffusion barriers. Dalili et al. [7] reported that amorphous $TaN_x$ films crystallized into a $Ta_2N$ structure at 600 °C when annealed in a 5% $H_2$–$N_2$ mixture for 30 min. Similarly, Uekubo et al. [14] stated that amorphous $WN_x$ films crystallized when annealed at 600 °C in 5% $H_2$–$N_2$ for 30 min. The annealing temperature for Cu to penetrate the diffusion barrier in a fixed time was correlated to the film thickness because of the diffusion-controlled process [7]. Suh et al. [15] reported that the as-deposited

100 nm-thick $WN_x$ films with 16%–32% N were X-ray amorphous, which crystallized after being annealed at 500–600 °C, and the barrier property was maintained up to 800 °C. Shen et al. [21] reported that 150 nm-thick amorphous $WN_x$ films crystallized above 600 °C when annealed in a vacuum for 30 min, and the crystallized $W_2N$ decomposed by releasing nitrogen above 820 °C. As-deposited $WN_x$ films could be either crystalline [22,23] or amorphous [21,23]. Si has been introduced into $WN_x$ films to change structures from crystalline to nanocomposite or amorphous, and the formed WSiN films exhibited improved mechanical properties [24,25], thermal stability [25], oxidation resistance [25–27], and corrosion resistance [28]. Thermal stability is a vital characteristic of the diffusion barrier. Louro and Cavaleiro [24] reported that amorphous WSiN films crystallized at 750–950 °C. Atomic layer deposition (ALD)-grown WSiN films were amorphous up to 800 °C and crystallized at 900 °C [29]. The diffusion of Cu into Si was prevented by a 6-nm-thick ALD-WSiN film when annealed in a vacuum at temperatures up to 600 °C for 30 min [29]. In contrast, a sputtered WSiN film with a thickness of 10 nm underwent 1 min annealing in $N_2$ up to 750 °C [30]. Chemical vapor deposition-prepared $W_{47}Si_9N_{44}$ films (100 nm thick) were useful barriers up to 700 °C [31]. In addition to the achievement of amorphous films, crystalline nitride films have also been applied as diffusion barriers [6]. Zhang et al. [32] reported that crystalline $W_2N$ films were stable up to 600 °C and decomposed at 800 °C in a vacuum. Takeyama and Noya [33] reported that a $W_{65}N_{35}$ film in the $W_2N$ phase exhibited barrier properties up to 800 °C for 1 h annealing. A 5-nm-thick TaWN film with a (111) orientation exhibited barrier properties against Cu diffusion at 500 °C in a vacuum for 1 h [34]. Crystalline CrWN films were applied as a diffusion barrier up to 650 °C in a vacuum for 1 h, as reported in a previous study [35]. In [36], a co-sputtered $W_{28}Si_{24}N_{48}$ film exhibited high oxidation resistance at 600 °C in an oxygen-containing atmosphere because of its amorphous structure and the formation of the $SiO_2$ scale, which implied its potential application for diffusion barriers. In this study, WSiN films were fabricated through hybrid pulse direct current/radio frequency magnetron co-sputtering. Crystalline and near-amorphous WSiN films with various Si contents were produced. The evolution in the crystalline structure and the variation in the diffusion barrier property caused by Si addition were investigated and correlated. The diffusion barrier characteristics of WSiN films that experienced annealing treatments in a vacuum at 500–900 °C for 1 h were evaluated.

## 2. Materials and Methods

WSiN films were co-sputtered with a W target (99.95%, 76.2 mm in diameter) linked to a pulse power supply and an Si target (99.999%, 50.8 mm in diameter) attached to a radio frequency (RF) power generator. The distance between targets and the substrate holder was 120 mm. The sputter guns were titled at an angle of 30° with respect to the substrate holder, which focused plasma toward the center of the substrate holder. The average power applied on W target ($P_W$) was fixed at 200 W, whereas the RF power on Si target ($P_{Si}$) varied from 0 to 50, 100, and 150 W, which fabricated samples S0, S50, S100, and S150, respectively, as Table 1 shows. The substrate holder was rotated at 10 rpm and maintained at 150 °C. The flow rates of Ar and $N_2$ gas were, respectively, set at 30 and 9 sccm for reactive sputtering. The working pressure was 0.4 Pa. Table 1 lists the experimental parameters for co-sputtering WSiN films. These samples were also designated by their chemical compositions examined using Auger electron spectroscopy (AES, PHI700, ULVAC-PHI, Kanagawa, Japan). The sputter etching rate for analyzing AES depth profiles was set at 9.1 or 9.5 nm/min for $SiO_2$. The chemical compositions of the WSiN films were determined as the average values related to depth ranges with sputter times of 11–25, 13–28, 15–30, and 10–20 min for samples S0, S50, S100, and S150, respectively, which exhibited $W_{76}N_{24}$, $W_{68}Si_0N_{32}$, $W_{63}Si_4N_{33}$, and $W_{59}Si_9N_{32}$, respectively, as the O content was ignored. These samples were laminated with a 107-nm-thick Cu layer prepared through direct current (DC) magnetron sputtering with a Cu target (99.99%, 76.2 mm in diameter) and DC power of 100 W for 18.5 min in a 30 sccm Ar flow and a working pressure of 0.4 Pa. Annealing experiments for evaluating the diffusion barrier characteristics were performed at 500–900 °C in a vacuum of $7 \times 10^{-4}$ Pa

for 1 h. The bonding characteristics of films were analyzed by using an X-ray photoelectron spectroscope (XPS, PHI 1600, PHI, Kanagawa, Japan). The C 1s line of the free surface of the $W_{76}N_{24}$ films was 283.97 eV. The C 1s line was calibrated using the Greczynski–Hultman method [37–39], which suggested a value of 0.18 eV for correcting binding energy levels in this study. The sputter etching rate in XPS analyses was 8.4 nm/min for $SiO_2$. An X-ray diffractometer (XRD, X'Pert PRO MPD, PANalytical, Almelo, The Netherlands) in the grazing incidence mode was used to analyze the phases. The lattice constants, $a_0$, of crystalline films were determined according to the following equation

$$a = a_0 + K \times \frac{\cos^2 \theta}{\sin \theta} \tag{1}$$

where $a$ is the lattice constant for the (111), (200), (220), and (311) reflection; $K$ is the constant; $\theta$ is the diffraction angle. The sheet resistance of the Cu/WSiN/Si samples was determined using a four-point probe [35]. The standard deviations for sheet resistance data were calculated from 3 measurements.

**Table 1.** Co-sputtering parameters of WSiN films.

| Sample | S0 | S50 | S100 | S150 |
|---|---|---|---|---|
| Average pulse power $P_W$ (W) | 200 | 200 | 200 | 200 |
| W target voltage (V) | 712 | 715 | 729 | 734 |
| W peak current (A) | 8.0 | 7.8 | 7.6 | 7.6 |
| W power density (kW/cm$^2$) | 0.22 | 0.22 | 0.22 | 0.22 |
| RF power $P_{Si}$ (W) | 0 | 50 | 100 | 150 |
| Thickness (nm) | 101 | 109 | 104 | 112 |
| Deposition time (min) | 29.0 | 21.8 | 20.1 | 18.4 |
| Deposition rate (nm/min) | 3.48 | 5.00 | 5.18 | 6.10 |
| Chemical composition | $W_{76}N_{24}$ | $W_{68}Si_0N_{32}$ | $W_{63}Si_4N_{33}$ | $W_{59}Si_9N_{32}$ |

## 3. Results and Discussion

### 3.1. Chemical Compositions and Phase Structures of As-Deposited W–Si–N Films

　　Figure 1 displays the AES depth profiles of the as-deposited Cu/WSiN/Si samples. The O content was contributed from the residual gas in the vacuum chamber. The relatively high O contents at the Cu/WSiN interfaces were attributed to the exposure of the chamber to the ambient environment for target changing from W to Cu. The chemical composition of the $WN_x$ films prepared without applying an RF power to the Si target ($P_{Si}$) was determined to be $W_{76}N_{24}$ after ignoring the O content. The WSiN films prepared with $P_{Si}$ levels of 50, 100, and 150 W exhibited chemical compositions of $W_{68}Si_0N_{32}$, $W_{63}Si_4N_{33}$, and $W_{59}Si_9N_{32}$, respectively. The Si content of the $W_{68}Si_0N_{32}$ films prepared with a $P_{Si}$ level of 50 W was not detectable. The N content of the WSiN films was maintained at a level of 32 at.%–33 at.%, which was close to the stoichiometric ratio of 2:1 for W:N in the $W_2N$ lattice [40]. The standard formation enthalpies for $W_2N$ and $Si_3N_4$ at 298 K are $-22$ [41] and $-745$ kJ/mol [42], respectively, which indicates a low affinity between N and W. Therefore, the $W_{76}N_{24}$ film exhibited a low N content. Moreover, the reactive co-sputtering with a Si target assisted in raising the N content in the films. However, the re-sputtering of light Si adatoms on the substrate affected the Si contents of the fabricated WSiN films [43]. The deposition rate increased from 3.48 to 5.00, 5.18, and 6.10 nm/min as the $P_{Si}$ was increased from 0 to 50, 100, and 150 W. The film thicknesses were fixed at 101–112 nm by controlling the deposition times (Table 1). Figure 2 exhibits the GIXRD patterns of the as-deposited WSiN films. The $W_{76}N_{24}$, $W_{68}Si_0N_{32}$, and $W_{63}Si_4N_{33}$ films formed a $W_2N$ phase (ICDD 00-025-1257), whereas the $W_{59}Si_9N_{32}$ film was nanocrystalline or near-amorphous. The dash lines in Figure 2 indicate the standard 2θ values of the $W_2N$ reflections (ICDD 00-025-1257) with a lattice constant of 0.4126 nm. The reflections of WSiN films shifted left with respect to the standard values of $W_2N$, implying expanded lattices for the WSiN films. The lattice constants of crystalline WSiN films were determined to be 0.4220, 0.4240, and 0.4229 nm

for the $W_{76}N_{24}$, $W_{68}Si_0N_{32}$, and $W_{63}Si_4N_{33}$ films, respectively. Similar peak shifts and lattice expansion were reported for the W–N films [21,22], which was attributed to excess nitrogen interstitials and intrinsic compressive stress [43]. The $W_{68}Si_0N_{32}$ films exhibited a higher lattice constant than the $W_{76}N_{24}$ films because of the N-deficient compositions for the $W_{76}N_{24}$ films, whereas the $W_{63}Si_4N_{33}$ films exhibited a lattice constant lower than that of the $W_{68}Si_0N_{32}$ films, which implied the substitution of small Si atoms replacing W atoms in the $W_2N$ lattice. Ju et al. [26] reported that the addition of Si into W–N films decreased the lattice constant; the films with a Si content < 5.2 at.% exhibited a $W_2N$ phase, whereas the films with a Si content in the range of 14.3 at.%–28.8 at.% were X-ray amorphous. The lattice shrinkage of these crystalline WSiN films after annealing is further discussed in Section 3.2.

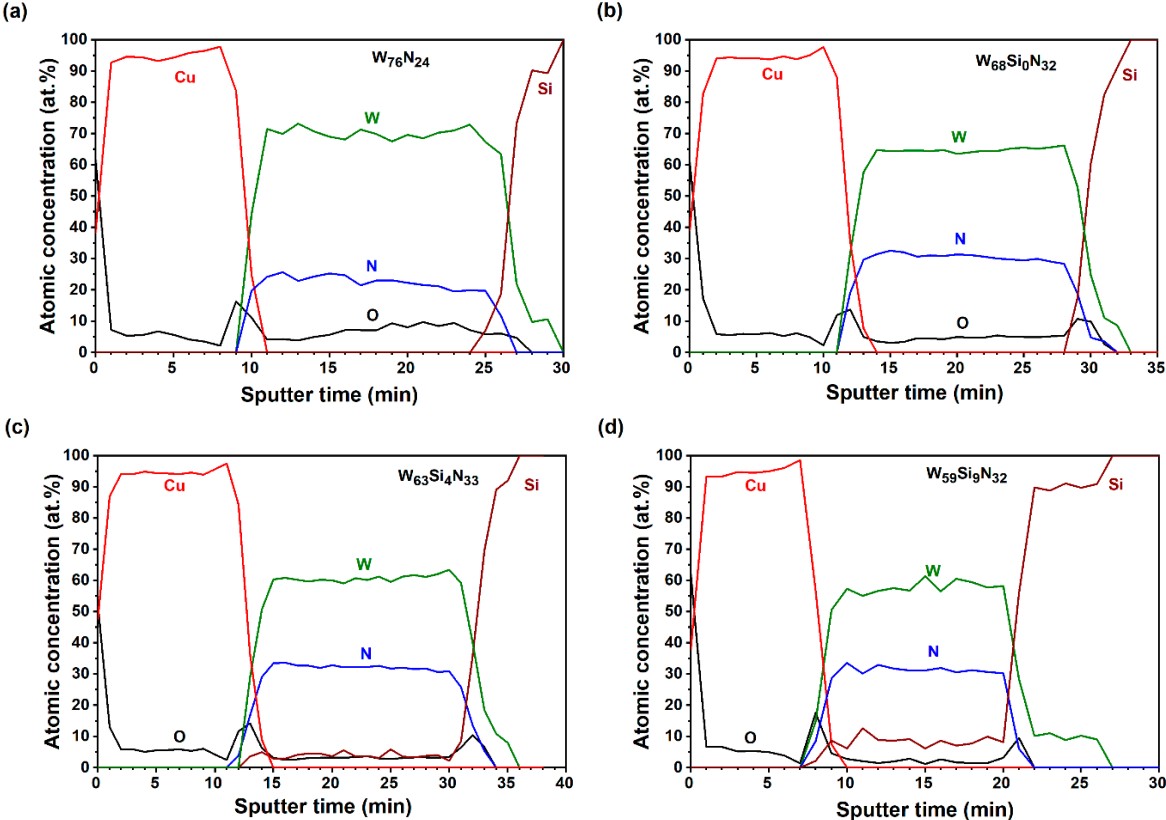

**Figure 1.** AES analyses of the as-deposited Cu/WSiN/Si samples with barrier layers: (**a**) $W_{76}N_{24}$, (**b**) $W_{68}Si_0N_{32}$, (**c**) $W_{63}Si_4N_{33}$, and (**d**) $W_{59}Si_9N_{32}$. (Sputter etching rates: 9.1 nm/min for (**b,c**) and 9.5 nm/min for (**a,d**).)

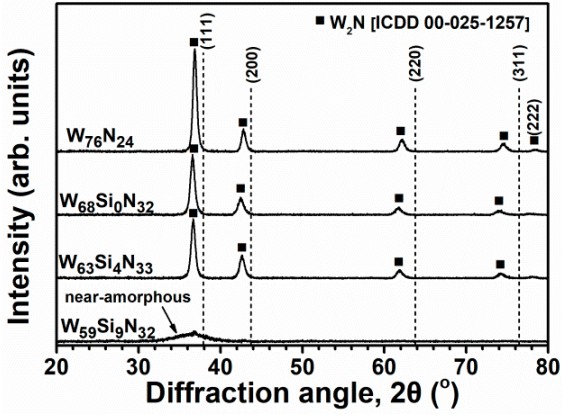

**Figure 2.** GIXRD patterns of the as-deposited WSiN films.

Figure 3 shows the XPS profiles at a depth of 33.6 nm of the as-deposited WSiN films without a Cu top layer. The profiles of the W 4f, Si 2p, and N 1s core levels of the aforementioned WSiN films are displayed. The W signals were split into two doublets, W–W and W–N, overlapped with a minor W $5p_{3/2}$ component (Figure 3a). Table 2 summarizes the XPS analysis results at depths of 8.4, 16.8, 25.2, and 33.6 nm. The binding energies of W $4f_{7/2}$ were 30.97–31.17 and 32.13–32.19 eV for the W–W and W–N bonds, respectively, which were comparable to the reported values of 31.26–31.52 and 32.36–32.56 eV [44], respectively. Figure 3b exhibits the Si 2p signals. The Si signal of the $W_{68}Si_0N_{32}$ films was not observed because of the low Si content, whereas the Si signals of the $W_{63}Si_4N_{33}$ and $W_{59}Si_9N_{32}$ films comprised two components for the Si–Si and Si–N bonds. The binding energies were 98.96–99.06 and 101.03–101.06 eV for the Si–Si and Si–N bonds, respectively, which were comparable with the reported values of 99.29–99.53 and 101.25–101.42 eV [44]. Figure 3c shows the N 1s signals, which comprise N–W and N–Si bonds of 397.20–397.65 and 396.84–396.88 eV, respectively.

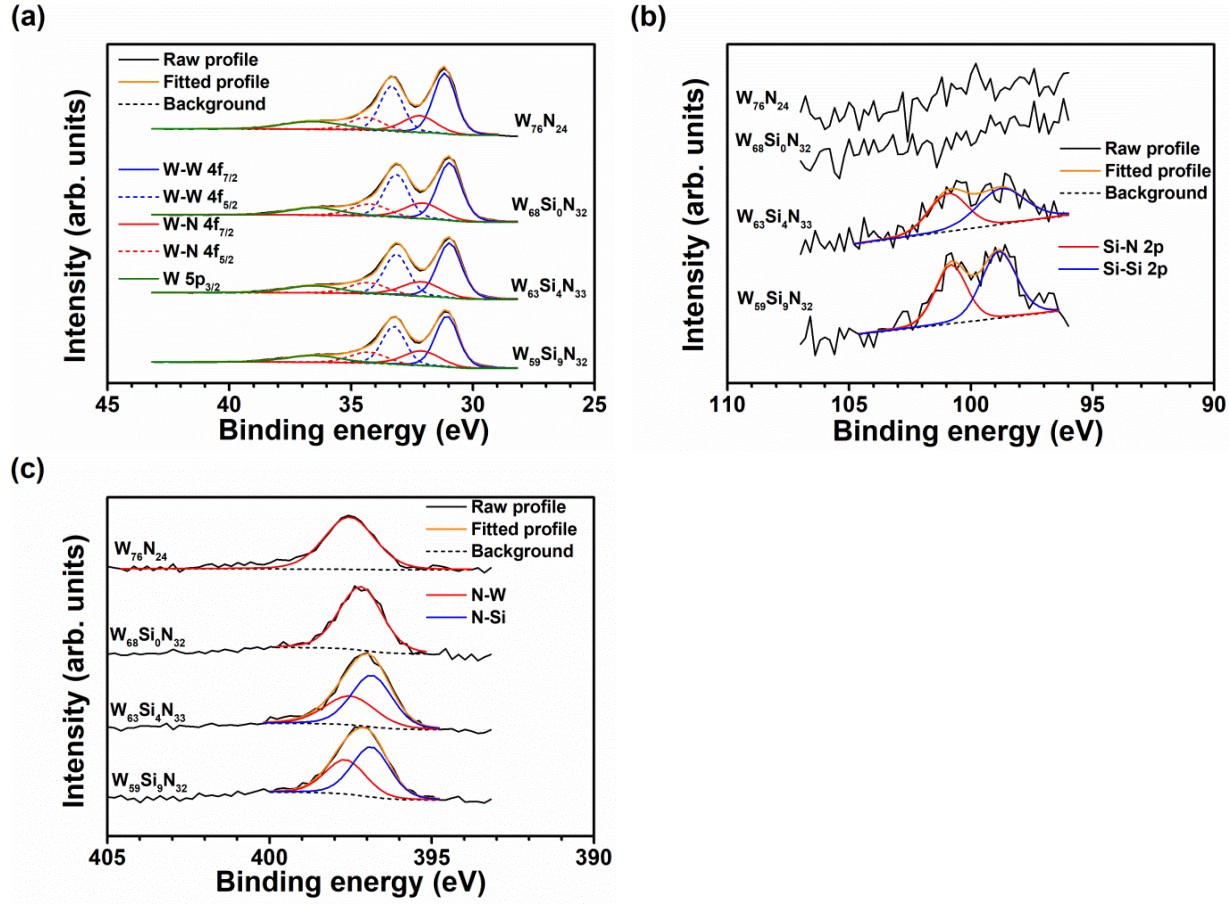

**Figure 3.** XPS patterns of (**a**) W 4f, (**b**) Si 2p, and (**c**) N 1s of the as-deposited WSiN films at a depth of 33.6 nm.

**Table 2.** XPS analysis results for the WSiN films at depths of 8.39–33.56 nm.

| Sample | W $4f_{7/2}$ (eV) | | Si 2p (eV) | | N 1s (eV) | |
|---|---|---|---|---|---|---|
| | W–W | W–N | Si–Si | Si–N | N–W | N–Si |
| $W_{76}N_{24}$ | 31.17 ± 0.03 | 32.19 ± 0.01 | - | - | 397.55 ± 0.01 | - |
| $W_{68}Si_0N_{32}$ | 30.98 ± 0.03 | 32.15 ± 0.02 | - | - | 397.20 ± 0.03 | - |
| $W_{63}Si_4N_{33}$ | 30.97 ± 0.02 | 32.14 ± 0.01 | 98.96 ± 0.04 | 101.06 ± 0.07 | 397.49 ± 0.09 | 396.84 ± 0.09 |
| $W_{59}Si_9N_{32}$ | 31.07 ± 0.01 | 32.13 ± 0.04 | 99.06 ± 0.04 | 101.03 ± 0.09 | 397.65 ± 0.05 | 396.88 ± 0.01 |

### 3.2. Structural Evolution after Annealing

Figure 4 exhibits the GIXRD patterns of the Cu/WSiN/Si samples after 1 h annealing in a vacuum at 500–900 °C. The samples with crystalline $W_{76}N_{24}$, $W_{68}Si_0N_{32}$, and $W_{63}Si_4N_{33}$ barrier films maintained Cu (ICDD 00-004-0836) and $W_2N$ phases after they were annealed up to 750 °C, and an extra $Cu_3Si$ (300) phase (ICDD 00-051-0916) was observed for the 800 °C-annealed samples. Moreover, these $W_2N$ reflections shifted toward higher 2θ values when increasing the annealing temperature, which resulted in the overlap of $W_2N$ (200) and Cu (111) reflections. Figure 5 shows the lattice constants of the aforementioned annealed WSiN films determined using (111), (220), and (311) reflections, which reveals a common decreasing tendency toward the standard lattice constant of the $W_2N$ phase when increasing the annealing temperatures. These originally expanded lattices shrunk after annealing, indicating that these crystalline WSiN films became more ordered after annealing. The low decreasing slope of the lattice constants of the $W_{63}Si_4N_{33}$ films with respect to annealing temperature was because of the formation of a certain volume of amorphous $SiN_x$. In contrast, the $Cu/W_{59}Si_9N_{32}/Si$ samples were maintained as near-amorphous after annealing up to 800 °C, as indicated by a weak and broad $W_2N$ (111) reflection in the XRD patterns. Moreover, a tiny W (110) reflection (ICDD 00-004-0806) was observed at annealing temperatures higher than 750 °C. Furthermore, a $Cu_3Si$ (300) reflection was observed for the 850 °C and 1 h-annealed $Cu/W_{59}Si_9N_{32}/Si$ sample. However, the $Cu_3Si$ (300) reflection was not shown for the XRD pattern of an 850 °C and 30 min-annealed $Cu/W_{59}Si_9N_{32}/Si$ sample. Figure 6 depicts the GIXRD pattern of the 900 °C-annealed $Cu/W_{59}Si_9N_{32}/Si$ sample, and Cu and $W_2N$ phases are replaced by $Cu_3Si$ and $WSi_2$ (ICDD 01-081-2168) phases.

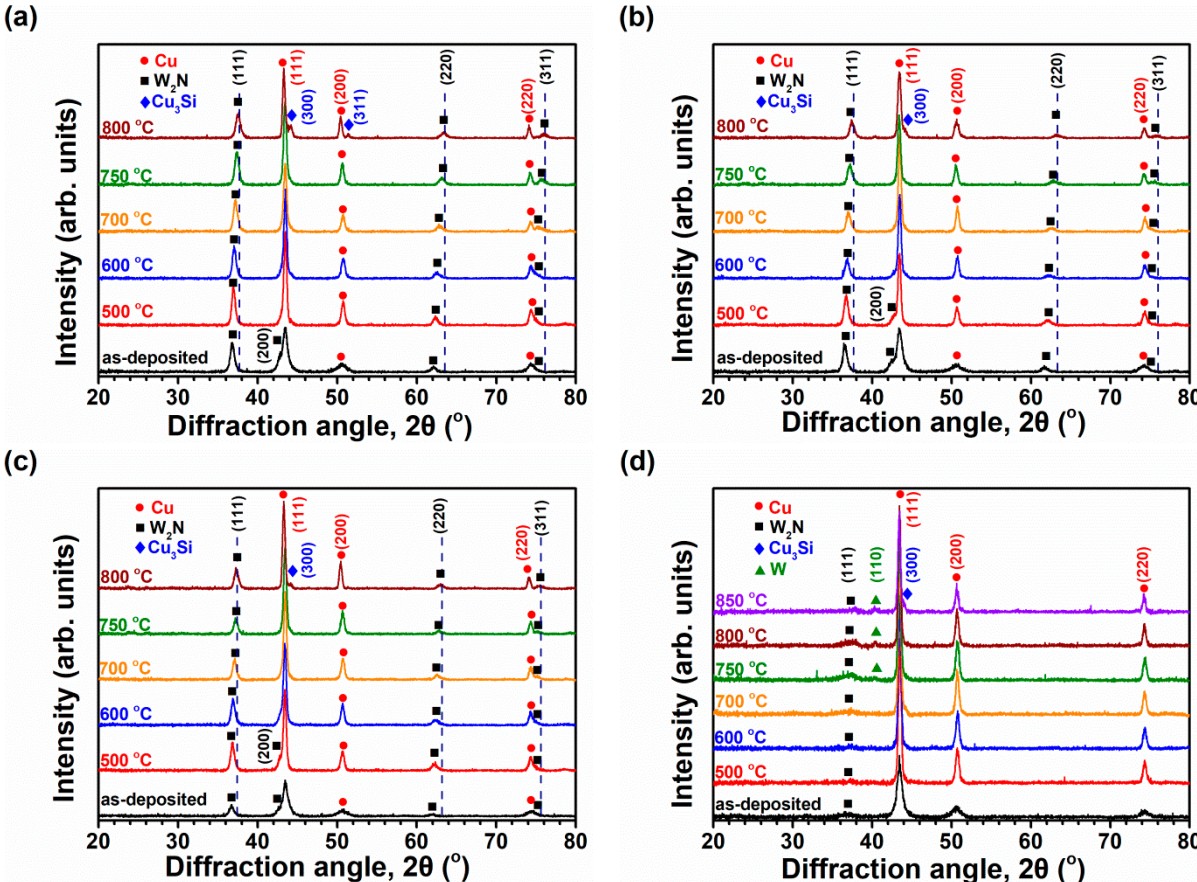

**Figure 4.** GIXRD patterns of the annealed Cu/WSiN/Si samples with barrier layers: (**a**) $W_{76}N_{24}$, (**b**) $W_{68}Si_0N_{32}$, (**c**) $W_{63}Si_4N_{33}$, and (**d**) $W_{59}Si_9N_{32}$.

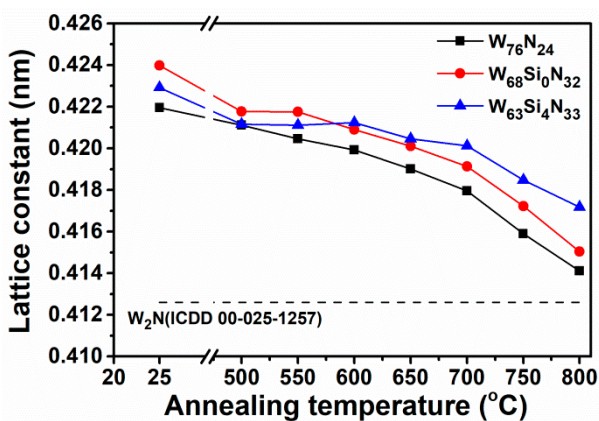

**Figure 5.** Lattice constant variations in the crystalline WSiN films after annealing.

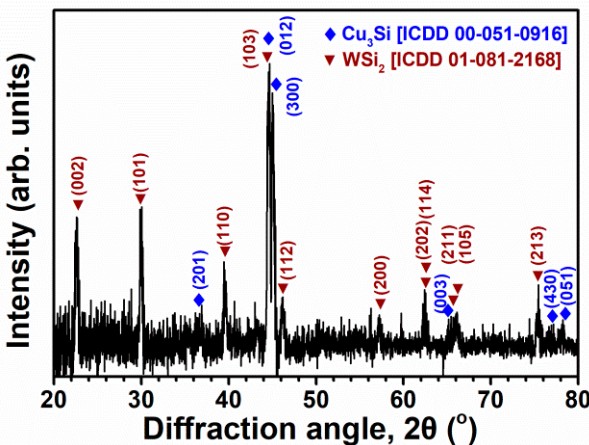

**Figure 6.** GIXRD pattern of the Cu/W$_{59}$Si$_9$N$_{32}$/Si sample after annealing at 900 °C for 1 h.

### 3.3. Diffusion Barrier Properties

Figure 7 displays the sheet resistance of the Cu/WSiN/Si samples at the as-deposited state and after annealing in a vacuum for 1 h at various temperatures. The sheet resistance of these Cu/WSiN/Si samples decreased from 0.9–1.2 Ω/□ to approximately 0.3 Ω/□ after they were annealed at 500 °C because of the defect annihilation and grain growth of Cu films [35,45], and then the sheet resistance of the aforementioned samples was maintained at the level of 0.3 Ω/□ up to 700 °C annealing. The sheet resistance of the aforementioned samples with crystalline WSiN barriers slightly increased to 0.4 Ω/□ when annealed at 750 °C and abruptly increased to 28–94 Ω/□ when annealed at 800 °C, which was accompanied by the formation of Cu$_3$Si (Figure 4). In contrast, the sheet resistance of the Cu/W$_{59}$Si$_9$N$_{32}$/Si sample was maintained at 0.3 Ω/□ up to 800 °C annealing, slightly increased to 0.4 Ω/□ after annealing at 850 °C, and abruptly increased to 5 Ω/□ after annealing at 900 °C.

Figure 8 shows the AES analysis results of the 650, 750, and 800 °C-annealed Cu/W$_{76}$N$_{24}$/Si samples. Correlated to the AES analysis results of the as-deposited Cu/W$_{76}$N$_{24}$/Si sample (Figure 1a), both the interdiffusion across the interfaces between Cu/W$_{76}$N$_{24}$ and W$_{76}$N$_{24}$/Si at 650 and 750 °C were limited; however, the formation of the Cu–Si compound did not happen, which agreed with that shown in the XRD pattern (Figure 4a). Further raising the annealing temperature to 800 °C resulted in the intermixing of the original Cu and W$_{76}$N$_{24}$ layers, the diffusion of Cu into the Si substrate, and the formation of a Cu$_3$Si phase as indicated in the XRD pattern, which was accompanied by a sharp increase in the sheet resistance of the Cu/W$_{76}$N$_{24}$/Si sample (Figure 7). Figure 9 displays the AES depth profiles of the 650, 800, 850, and 900 °C-annealed Cu/W$_{59}$Si$_9$N$_{32}$/Si samples. Both the interdiffusion behaviors across the interfaces between Cu/W$_{59}$Si$_9$N$_{32}$ and W$_{59}$Si$_9$N$_{32}$/Si

at 650–800 °C were restricted. The intermixing of the original Cu and $W_{59}Si_9N_{32}$ layers occurred after 850 °C annealing. Because Cu and W were immiscible and Cu and N formed no compound, the newborn phase formed due to the intermixing of Cu and $W_{59}Si_9N_{32}$ layers was $Cu_3Si$, as shown in the XRD pattern (Figure 4d); however, the amount of $Cu_3Si$ phase was little, and the sheet resistance increased from 0.3 Ω/□ for samples annealed below 800 °C to 0.4 Ω/□ for the sample annealed at 850 °C. Moreover, the nitrogen-loss behavior was observed above 850 °C. Shen et al. [21] reported that the $W_2N$ phase was stable up to 800 °C and started to evaporate at 820 °C in a vacuum. Further raising the annealing temperature to 900 °C resulted in the inward diffusion of Cu into the Si substrate, the disappearance of the Cu layer, the outward diffusion of Si, the formation of $Cu_3Si$ and $WSi_2$ phases, and an increase in sheet resistance to 5 Ω/□. The sheet resistance measurement and AES analyses confirmed the diffusion barrier characteristics for both the crystalline and near-amorphous WSiN films.

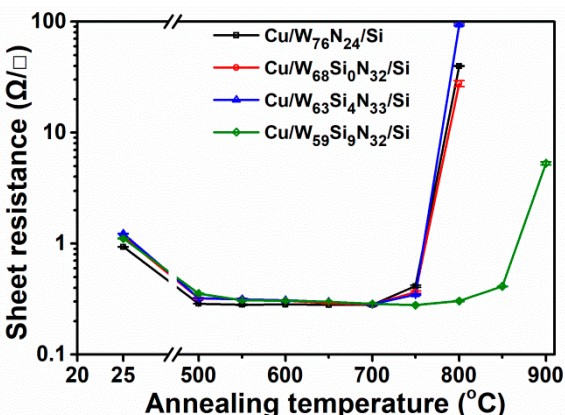

**Figure 7.** Sheet resistance of the annealed Cu/WSiN/Si samples.

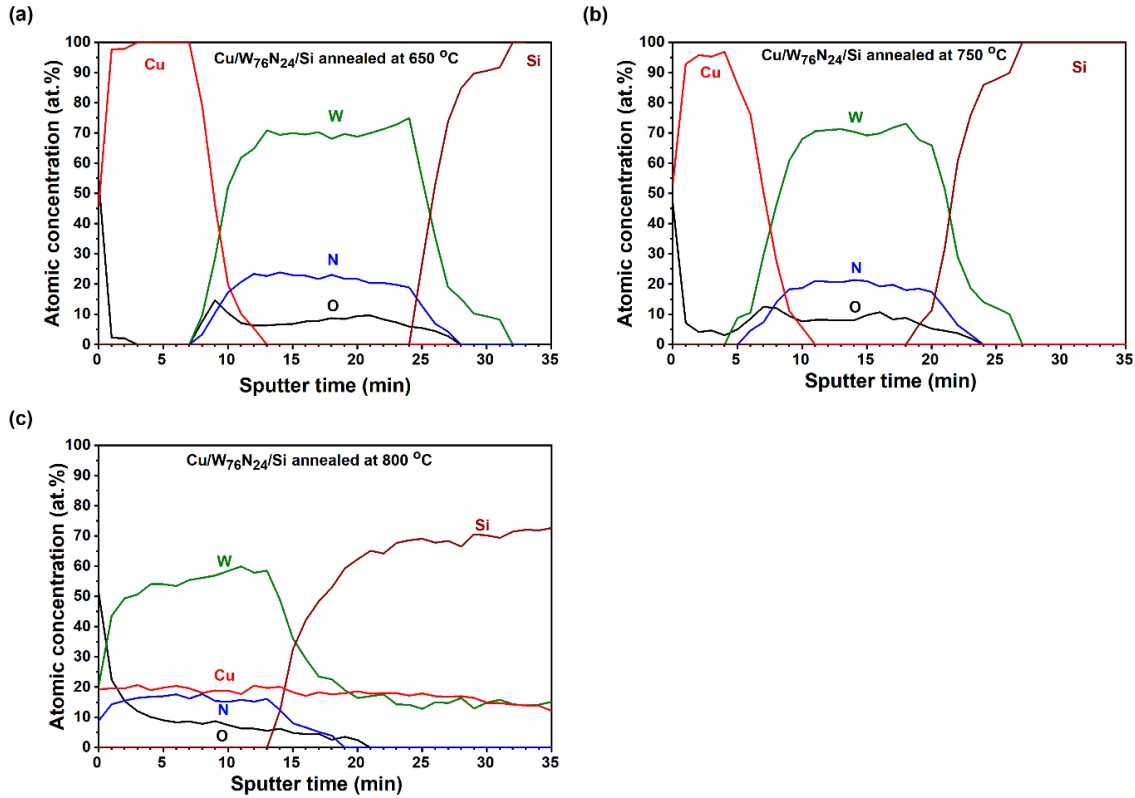

**Figure 8.** AES analysis results of (**a**) 650 °C-, (**b**) 750 °C-, and (**c**) 800 °C-annealed $Cu/W_{76}N_{24}/Si$ samples. (Sputter etching rate: 9.5 nm/min.)

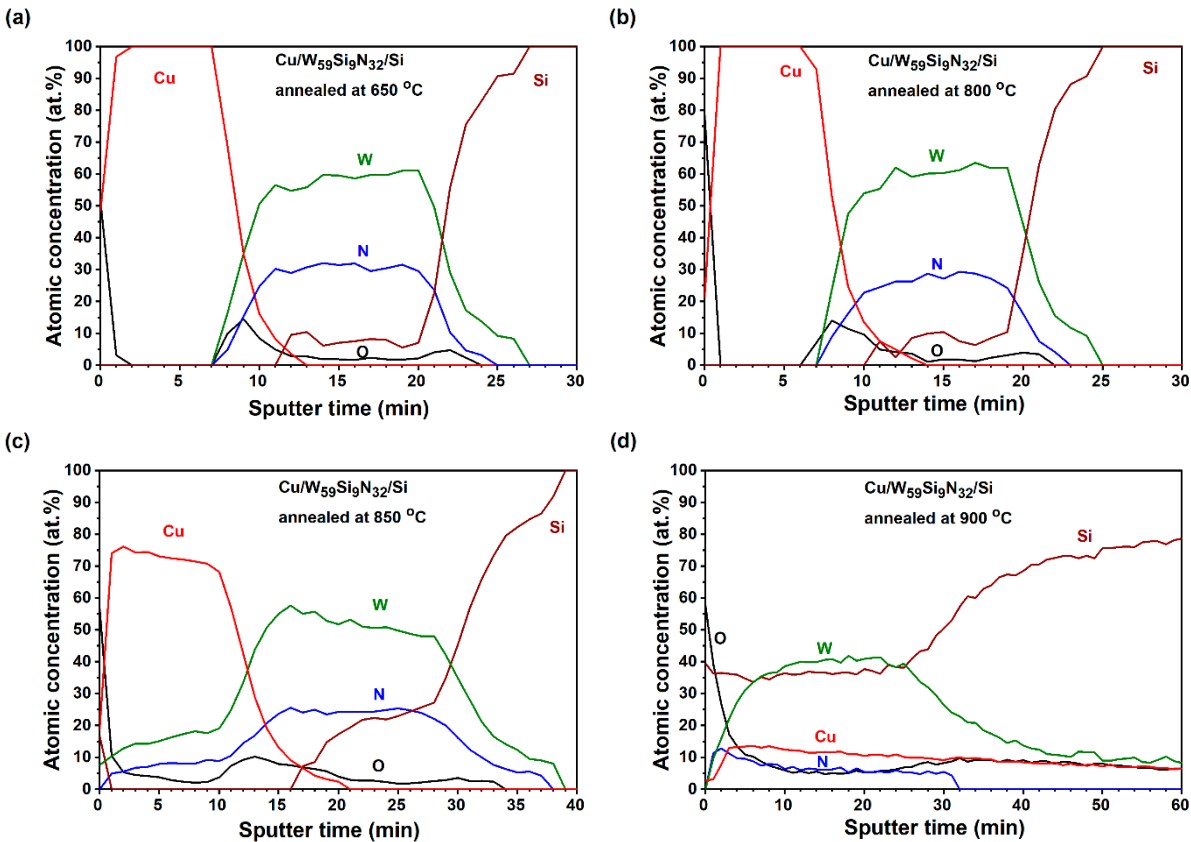

**Figure 9.** AES analysis results of (**a**) 650 °C-, (**b**) 800 °C-, (**c**) 850 °C-, and (**d**) 900 °C-annealed Cu/$W_{59}Si_9N_{32}$/Si samples. (Sputter etching rates: 9.1 nm/min for (**c,d**) and 9.5 nm/min for (**a,b**).)

## 4. Conclusions

The feasibility of WSiN films as diffusion barriers for Cu metallization was explored in this study. Both crystalline and near-amorphous WSiN films behaved as excellent diffusion barriers. The crystalline WSiN films with Si contents of 0 at.%–4 at.% exhibited diffusion barrier characteristics after annealing up to 750 °C in a vacuum for 1 h. The formation of the $Cu_3Si$ phase observed from XRD patterns, the inward diffusion of Cu through the films into Si substrates verified by AES analyses, and the sharp increase in sheet resistance indicated the failure of these crystalline WSiN barrier films when they were annealed at 800 °C for 1 h. The diffusion barrier characteristics of the WSiN films were further improved by raising the Si content to 9 at.%, which formed a near-amorphous structure in the as-deposited state and increased the failure temperature against Cu diffusion to 900 °C. Further research on shrinking the depth of WSiN films to the nano-scale accompanied by compatible diffusion barrier characteristics is crucial.

**Author Contributions:** Conceptualization, Y.-I.C. and L.-C.C.; validation, L.-C.C.; formal analysis, K.-H.Y.; investigation, K.-H.Y. and T.-Y.O.; resources, L.-C.C.; data curation, Y.-I.C.; writing—original draft preparation, Y.-I.C.; writing—review and editing, L.-C.C.; project administration, Y.-I.C. and L.-C.C.; funding acquisition, Y.-I.C. and L.-C.C. All authors have read and agreed to the published version of the manuscript.

**Funding:** This research was funded by the Ministry of Science and Technology, Taiwan, grant numbers 110-2221-E-131-013 and 110-2221-E-019-015. The APC was funded by Ming Chi University of Technology.

**Institutional Review Board Statement:** Not applicable.

**Informed Consent Statement:** Not applicable.

**Data Availability Statement:** Not applicable.

**Acknowledgments:** The analysis support from the Instrumentation Center at the National Tsing Hua University for the AES, XPS, and EPMA characterizations is acknowledged.

**Conflicts of Interest:** The authors declare no conflict of interest.

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
