# Peer review of "Diffusion Barrier Characteristics of WSiN Films"

_coatings, doi:10.3390/coatings12060811_

Round 1

Reviewer 1 Report

The research work is well presented and the results clearly and extensively discussed. Just few small comments:

- In the Introduction section the novelty of the work could be highlighted mainly at the end of the paragraph.

- Materials and Methods. Some more details about the targets? Why have they different diameter? Which is their angle respect to the sample holder?

- Conclusions could be improved putting more in lights the outcomes of the work also in relation to the future potential applications

Author Response

- In the Introduction section the novelty of the work could be highlighted mainly at the end of the paragraph.

A: Related illustrations were revised as “In [36], a co-sputtered W28Si24N48 film exhibited high oxidation resistance at 600 °C in an oxygen-containing atmosphere because of its amorphous structure and the formation of the SiO2 scale, which implied its potential application for diffusion barriers. In this study, WSiN films were fabricated through hybrid pulse direct current/radio frequency magnetron co-sputtering. Crystalline and near-amorphous WSiN films with various Si contents were produced. The evolution in crystalline structure and variation in diffusion barrier property caused by Si addition were investigated and correlated. The diffusion barrier characteristics of WSiN films that experienced annealing treatments in a vacuum at 500–900 °C for 1 h were evaluated.” in lines 73–82.

- Materials and Methods. Some more details about the targets? Why have they different diameter? Which is their angle respect to the sample holder?

A: New sentences “The distance between targets and the substrate holder was 120 mm. The sputter guns were titled at an angle of 30° with respect to the substrate holder, which focused plasma toward the center of the substrate holder.” were added in lines 86–88.

- Conclusions could be improved putting more in lights the outcomes of the work also in relation to the future potential applications

A: In lines 250–252, “The sheet resistance measurement and AES analyses confirmed the diffusion barrier characteristics for both the crystalline and near-amorphous WSiN films.” was added.

Conclusions were modified. New sentences were added. They were “Both crystalline and near-amorphous WSiN films behaved as excellent diffusion barriers.” and “Further research on shrinking the depth of WSiN films to the nano-scale accompanied by compatible diffusion barrier characteristics is crucial.”

Reviewer 2 Report

The reviewed manuscript investigates the structural evolution and the variation in diffusion barrier property caused by Si addition to the crystalline and near-amorphous WSiN films. The article is made at a good scientific and technical level, and its practical significance is beyond doubt. In order to improve the readability and clarity of the manuscript, some major concerns need to be addressed before the paper is to be accepted for publishing:

1. L 54: “…… amorphous [22–24]. Si has been introduced into WNx films to vary structures from crystalline to nanocomposite or amorphous [25–29] …” Try to avoid lumped references; a short comment should be included for each reference or two references in the same subject.

2. L 58 & L 62: “ALD-grown & CVD-prepared” please define any abbreviations for the first time appear before using in the text.

3. The motivation for the study and the research gap are not clear enough. Please demonstrate in the introduction of the paper, the novelty of this research in relation to other thematically similar research papers.

4. The literature review is quite out of date, only 4-5 new ref out of 40. please improve it with novel references.

5. Table 1: There is no Si content in both samples (W76N24) and (W68Si0N32), Why it is written in different forms showing Si0 in the second sample only?

6. Please explain in the experimental section how the atomic percentage was calculated and adjusted to produce the selected samples composition. If the key parameter is “RF power Psi”, then it is recommended to relate the samples names to the given power. Ex: P0, P50, P100 and P150. In this case, the chemical composition (atomic percentage) will be presented as a result to the selected specific power.

7. The author needs to supplement the EDS analysis to confirm the Si percentages generated each sample, since an acceptable evidence is required as a proof.

8. Figure 2, GIXRD: Remove the intensity values from Y-axis, since there are more than pattern in the same graph.

9. L 119: “The dash lines in Figure 2 indicated the standard 2θ values of the W2N reflections [ICDD 00-025-1257]” Please explain or attribute to a clear scientific reason, why that “peak shift” is there?

10. Figure 2, GIXRD: Please provide the ICDD references for each phase in the pattern and identify all available peaks.

11. Figure 2, GIXRD: Where are the Si peaks in the 3rd and 4th samples W63Si4N33 and W59Si9N32 respectively? Note that, the Si main peak is before 2thata of 30 degrees either it is crystalline, polycrystalline or amorphous.

12. Discussion is lake of scientific explanation for the obtained results. Authors should attribute the results achieved to a clear scientific reason.

13. The English language used in the paper is to be revised and improved before the subsequent manuscript submission. Please, read the text carefully before the next submission of the paper.

Author Response

  1. L 54: “…… amorphous [22–24]. Si has been introduced into WNx films to vary structures from crystalline to nanocomposite or amorphous [25–29] …” Try to avoid lumped references; a short comment should be included for each reference or two references in the same subject.

A: Related illustrations were modified as:

Lines 41–43: “High-entropy alloy nitride films, such as (AlMoNbSiTaTiVZr)50N50 [10], (AlCrTaTiZr)N [18], and (AlCrTaTiZrMo)N [19], with amorphous structures and large lattice distortions, have been applied as diffusion barrier [20].”

Lines 55–56: “As-deposited WNx films could be either crystalline [22,23] or amorphous [21,23].”

Line 56–59: “Si has been introduced into WNx films to change structures from crystalline to nanocomposite or amorphous, and the formed WSiN films exhibited improved mechanical properties [24,25], thermal stability [25], oxidation resistance [25–27], and corrosion resistance [28].”

  1. L 58 & L 62: “ALD-grown & CVD-prepared” please define any abbreviations for the first time appear before using in the text.

A: Thanks for the reminding. “Atomic-layer-deposition (ALD)” and “Chemical-vapor-deposition” were corrected in lines 61 and 65, respectively.

  1. The motivation for the study and the research gap are not clear enough. Please demonstrate in the introduction of the paper, the novelty of this research in relation to other thematically similar research papers.

A: Related illustrations were revised as “In this study, WSiN films were fabricated through hybrid pulse direct current/radio frequency magnetron co-sputtering. Crystalline and near-amorphous WSiN films with various Si contents were produced. The evolution in crystalline structure and variation in diffusion barrier property caused by Si addition were investigated and correlated. The diffusion barrier characteristics of WSiN films that experienced annealing treatments in a vacuum at 500–900 °C for 1 h were evaluated.” in lines 76–82.

  1. The literature review is quite out of date, only 4-5 new ref out of 40. please improve it with novel references.

A: There are 20 references published in the last 10 years (2012-2022). The authors believe that the readers acquire complete references related to this topic.

  1. Table 1: There is no Si content in both samples (W76N24) and (W68Si0N32), Why it is written in different forms showing Si0in the second sample only?

A: As mentioned in lines 130–131, “The Si content of the W68Si0N32 films prepared with a PSi level of 50 W was not detectable.” The new sentence “However, the re-sputtering of light Si adatoms on the substrate affected the Si contents of the fabricated WSiN films [43].” was added in lines 136–137.

  1. Please explain in the experimental section how the atomic percentage was calculated and adjusted to produce the selected samples composition. If the key parameter is “RF power Psi”, then it is recommended to relate the samples names to the given power. Ex: P0, P50, P100 and P150. In this case, the chemical composition (atomic percentage) will be presented as a result to the selected specific power.

A: Thanks for the suggestion. The samples were named as S0, S50, S100, and S150 as listed in Table 1. The new sentences were added.

Lines 88–91: “The average power applied on W target (PW) was fixed at 200 W, whereas the RF power on Si target (PSi) varied from 0 to 50, 100, and 150 W, which fabricated samples S0, S50, S100, and S150, respectively, as Table 1 shows.”

Lines 97–101: “The chemical compositions of the WSiN films were determined as the average values related to depth ranges with sputter times of 11–25, 13–28, 15–30, and 10–20 min for samples S0, S50, S100, and S150, respectively, which exhibited W76N24, W68Si0N32, W63Si4N33, and W59Si9N32, respectively, as the O content was ignored.”

  1. The author needs to supplement the EDS analysis to confirm the Si percentages generated each sample, since an acceptable evidence is required as a proof.

A: The WSiN films were prepared on Si substrates. EDS analysis could not provide accurate Si content for these samples.

  1. Figure 2, GIXRD: Remove the intensity values from Y-axis, since there are more than pattern in the same graph.

A: The intensity values from Y-axis were removed.

  1. L 119: “The dash lines in Figure 2 indicated the standard 2θ values of the W2N reflections [ICDD 00-025-1257]” Please explain or attribute to a clear scientific reason, why that “peak shift” is there?

A: New sentences “The reflections of WSiN films shifted left with respect to the standard values of W2N, implying expanded lattices for the WSiN films.” and “Similar peak shifts and lattice expansion were reported for W–N films [21,22], which was attributed to excess nitrogen interstitials and intrinsic compressive stress [43].” were added in lines 144–150.

  1. Figure 2, GIXRD: Please provide the ICDD references for each phase in the pattern and identify all available peaks.

A: Figures 2 & 6 were corrected. All the ICDD references were provided.

  1. Figure 2, GIXRD: Where are the Si peaks in the 3rdand 4thsamples W63Si4N33 and W59Si9N32 respectively? Note that, the Si main peak is before 2thata of 30 degrees either it is crystalline, polycrystalline or amorphous.

A: Because Si atoms substituted into the W2N lattice in W63Si4N33 and the W59Si9N32 film was nanocrystalline or near-amorphous, no Si peaks were observed.

  1. Discussion is lake of scientific explanation for the obtained results. Authors should attribute the results achieved to a clear scientific reason.

A: New sentences were added.

Line 133–137 : “The standard formation enthalpies for W2N and Si3N4 at 298 K are −22 [41] and −745 kJ/mol [42], respectively, which indicates a low affinity between N and W. Therefore, the W76N24 film exhibited a low N content. Moreover, the reactive co-sputtering with a Si target assisted in raising the N content in the films. However, the re-sputtering of light Si adatoms on the substrate affected the Si contents of the fabricated WSiN films [43].”

Lines 153–156: “Ju et al. [26] reported that the addition of Si into WN films decreased the lattice constant; the films with a Si content <5.2 at.% exhibited a W2N phase , whereas the films with a Si content in the range of 14.3–28.8 at.% were X-ray amorphous.”

  1. The English language used in the paper is to be revised and improved before the subsequent manuscript submission. Please, read the text carefully before the next submission of the paper.

A: The revised manuscript was edited again by MDPI.

Reviewer 3 Report

The authors present an extensive study exploring the feasibility of co-sputtered WSiN films as diffusion barriers for Cu metallization. The structural properties of deposited and annealed films were comprehensively studied by means of XPS, GIXRD, Auger electron spectroscopy. The research is well designed, the results seem to be correct and are important for the future development of ultra-large-scale integrated circuits.
However, the text can be improved, I have few comments:
1) The error bars are not presented in the Fig.5 and Fig.7. The value of error is not described in the text.
2) The motivation of studying the WSiN thin films as diffusion barriers for Cu is not described in Introduction, can be improved.

Author Response

1) The error bars are not presented in the Fig.5 and Fig.7. The value of error is not described in the text.

A: “The lattice constants, É‘0, of crystalline films were determined according to the following equation:

(Equation) (1)

where É‘ is the lattice constant for the (111), (200), (220), and (311) reflection; K is the constant; and θ is the diffraction angle.” was added in lines 112–115. No error bars are presented for Fig. 5.

Fig.7 was revised by adding the error bars. New sentences “The sheet resistance of the Cu/WSiN/Si samples was determined using a four-point probe [35]. The standard deviations for sheet resistance data were calculated from 3 measurements.” were added in lines 115–117.

2) The motivation of studying the WSiN thin films as diffusion barriers for Cu is not described in Introduction, can be improved.

A: Related illustrations were revised as “In [36], a co-sputtered W28Si24N48 film exhibited high oxidation resistance at 600 °C in an oxygen-containing atmosphere because of its amorphous structure and the formation of the SiO2 scale, which implied its potential application for diffusion barriers. In this study, WSiN films were fabricated through hybrid pulse direct current/radio frequency magnetron co-sputtering. Crystalline and near-amorphous WSiN films with various Si contents were produced. The evolution in crystalline structure and variation in diffusion barrier property caused by Si addition were investigated and correlated. The diffusion barrier characteristics of WSiN films that experienced annealing treatments in a vacuum at 500–900 °C for 1 h were evaluated.” in lines 73–82.

Round 2

Reviewer 2 Report

     The revision is satisfactory and the authors have provided amendments to all the suggested queries. Therefore, I recommend this work for publication in Coatings Journal.